# Application of the YOLOv6 Combining CBAM and CIoU in Forest Fire and Smoke Detection



Aoran Wang †, Guanghao Liang, Xuan Wang † and Yongchao Song *

School of Computer and Control Engineering, Yantai University, Yantai 264005, China;
aoranwang@s.ytu.edu.cn (A.W.); 202158505127@s.ytu.edu.cn (G.L.); xuanwang91@ytu.edu.cn (X.W.)
* Correspondence: ycsong@ytu.edu.cn
† These authors contributed equally to this work.

**Abstract:** Forest fires are a vulnerable and devastating disaster that pose a major threat to human property and life. Smoke is easier to detect than flames due to the vastness of the wildland scene and the obscuring vegetation. However, the shape of wind-blown smoke is constantly changing, and the color of smoke varies greatly from one combustion chamber to another. Therefore, the widely used sensor-based smoke and fire detection systems have the disadvantages of untimely detection and a high false detection rate in the middle of an open environment. Deep learning-based smoke and fire object detection can recognize objects in the form of video streams and images in milliseconds. To this end, this paper innovatively employs CBAM based on YOLOv6 to increase the extraction of smoke and fire features. In addition, the CIoU loss function was used to ensure that training time is reduced while extracting the feature effects. Automatic mixed-accuracy training is used to train the model. The proposed model has been validated on a self-built dataset containing multiple scenes. The experiments demonstrated that our model has a high response speed and accuracy in real-field smoke and fire detection, which provides intelligent support for forest fire safety work in social life.

**Keywords:** forest fire and smoke detection; YOLOv6; CBAM; feature enhancement





## 1. Introduction

Forest fires severely damage the environment and also seriously jeopardize the safety of people's lives and property. To minimize the damage, this requires that forest fires be detected as soon as they occur. In the case of wildfires, smoke is often more visible than fire due to the influence of vegetation. Therefore, it is important to detect fires and smoke in the wild. Due to the spatial and temporal uncertainty of smoke and fire events, it has been difficult for traditional manual monitoring methods and high-cost server-side-based image recognition to meet the current needs, while the use of hardware to inspect specific areas suffers from high-cost design, small detection area, and slow response time. Therefore, how to solve the problem of specific object detection and timely response to smoke and fires, reducing the cost and improving the detection accuracy to reduce the rate of false alarms, is currently an urgently needed solution.

Deep learning has a wide range of applications in the field of computer vision, in which object detection is an important research direction. In many real-world scenarios, such as fires, smoke, and other catastrophic events, being able to quickly and accurately detect fire sources and smoke areas is crucial for rescue work. Therefore, deep learning-based smoke and fire object detection is of great research importance.

Currently, research for smoke and fire object detection is mainly based on traditional computer vision algorithms and deep learning methods. Traditional computer vision algorithms can realize the task of object detection to a certain extent, but they still have many limitations for the processing of complex scenes. For example, traditional sensor-based methods [1–3] are susceptible to interference factors such as spatial dimensions. Its diffusivity leads to reduced smoke concentration, with limitations such as limited detection

range and delayed alarm time [4]. Chen et al. [5] proposed the use of two smoke sensors, two fire sensors, and two temperature sensors to detect fire events. But this method still faces the problem that in open areas, smoke and fire information is diluted by the environment and difficult to detect in time.

Deep learning-based methods can utilize large amounts of data and high-performance computational resources, which can better handle object detection tasks in complex scenarios and achieve better results. For example, Wang et al. [4] applied YOLOv5 to achieve fire and smoke recognition. Jia et al. [6] used CNN to recognize fire smoke. He et al. [7] improved CNN by designing an attention mechanism that combines spatial and channel attention, and a decision fusion module was used to differentiate smoke and fire. For better recognition, Pan et al. [8], Li et al. [9] and Wu et al. [10] used Faster RCNN, Efficientdet and SSD to extract smoke static features respectively.

Although deep learning-based object detection models are better than traditional models, they still have many problems when dealing with complex environments outdoors. For example, in cloudy weather in the evening, the fire and smoke effects created by the sunset and clouds are easily misjudged as smoke and fire. Especially when it is cloudy and rainy in the forest, the black clouds in the sky are often misreported as smoke. These factors will greatly affect the object detection accuracy.

Superior feature extraction networks can extract as many desired features as possible to better solve the above problems. Compared to traditional CNN feature extraction networks, FPN [11] and its variants further enhance feature extraction and are widely used in deep learning network models [12–14]. The variants of FPN are PANet [15], NAS-FPN [16], and BiFPN [17]. In addition, in object detection tasks, the irregularity of the object also affects the detection results. Therefore, a feature extraction network with a rotation module helps to improve multi-scale feature fusion capability.

In addition to feature extraction, the attention mechanism and loss function also have an impact on detection accuracy. The attention mechanism is essentially a resource allocation mechanism which can change the method of resource allocation according to the degree of importance of the object of attention so that the resources are more inclined to the object of attention. In convolutional neural networks, the resource to be allocated by the attention mechanism is the weight parameter. Assigning more weight parameters to the attention object can enhance the feature extraction of the attention object during the model training process. Adding the attention mechanism to the object detection task can improve the model's characterization ability, effectively reducing the interference of invalid objects, improving the detection effect on the attention object, thus improving the overall detection accuracy of the model. The loss function is used mainly in the training phase of the model. After each batch of training data is fed into the model, the predicted value is output through forward propagation, and then the loss function calculates the difference between the predicted value and the true value, which is the loss value. After obtaining the loss value, the model updates each parameter through backpropagation to reduce the loss between the true value and the predicted value, so that the predicted value generated by the model is closer to the true value, thus achieving the purpose of learning.

To address the above problems, YOLOv6 [18] is selected as the benchmark model in this paper. It adopts the Rep-PAN feature extraction network, replacing the CSP-Block used in YOLOv5 with RepBlock, and also adjusts the operators in the overall neck. We added the CBAM attention module to YOLOv6 and replaced the original GIoU [19] loss function with the CIoU [20] loss function.

Our major research efforts are as follows:

- Most of the existing pyrotechnic detection techniques in the YOLO series use YOLOv5 as the benchmark model. To verify the performance of other techniques, we innovatively choose YOLOv6 to be the baseline model.
- Based on the original model, we introduce the CBAM attention mechanism so that the model achieves efficient inference in hardware while maintaining a better multi-scale feature fusion capability. We use CIoU as the loss function of the model as a way to

obtain higher detection accuracy. In addition, we added an automatic mixed-precision AMP when training the model. It can be calculated with different data precision for different layers in the neural network inference process, thus realizing the purpose of saving video memory and speeding up the process. The detection accuracy of the model is further improved.

- We collected part of the public firework dataset independently and supplemented it with other datasets that were labeled. After data cleaning, we produced high-quality datasets. We conducted experiments on our data for comparison and validation. The final experimental results prove the merits of the model in this paper.

The rest of the paper is organized as follows. Section 2 describes the data sources in detail, and in Section 3, we detail our improved methodology. Our experimental configuration and results are shown in Section 4. In Section 5, we discuss prospective future research and challenges. In the last section, the whole paper is concluded.

## 2. Datasets

In order to better carry out the experimental study in this paper, we carried out data collection and organization, called WFDS2000. The preparation of this dataset collection was carried out by web crawling a partially publicly labeled firework dataset to obtain a total of about 3000 pictures. Then it was divided into a training set and a validation set according to a ratio of 7:3, and some of the datasets were relabeled. This allowed the smoke and fire objects to be cleaned by framing the dataset in as large a box as possible, discarding the many smaller boxes of the labeling method. Ensuring the quality of the dataset is a better place to start. There are many cases of overfitting or failure of the model to converge, the ultimate cause of which is a contaminated dataset. Figure 1 shows some example images from WFDS2000. WFDS2000 contains multi-scene images such as field and city. In addition, WFDS2000 contains multiple smoke types such as white smoke, black smoke, and thin smoke. In Table 1, we present the details of the dataset. It contains the number of images and the specific number of pyrotechnic objects in the training set, validation set, and test set.

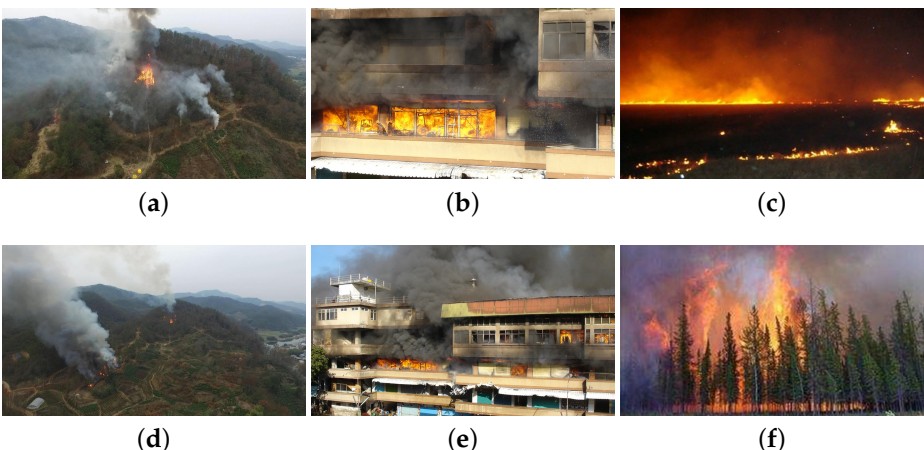

**Figure 1.** Images of WFDS2000 for different scenarios. (**a**–**f**) are partial display images in WFDS2000.

**Table 1.** Specific information about WFDS2000.

| Datasets | Number of Fire | Number of Smoke | Image Amount |
|---|---|---|---|
| train | 2015 | 1822 | 1576 |
| test | 688 | 597 | 525 |
| val | 632 | 625 | 525 |

## 3. Methods

Existing methods tend to have high false alarm rates when confronted with interference from other factors, such as clouds and smoke. In addition, the irregularity of fire and smoke further affects the detection performance. Therefore, unlike the commonly used YOLOv5-based object detection methods, we choose the YOLOv6 model with a rotating detection head as the baseline model. For the interference of other factors, we add CBAM. Finally, we choose the appropriate loss function CIoU to further enhance the performance of the model and supplement it with AMP automatic mixed-accuracy training.

### 3.1. Excellent Network Design

The YOLO series usually consists of backbone, neck, and head. In particular, the performance of the backbone determines the feature representation capability. At the same time, its design has a crucial impact on inference efficiency because it bears a large amount of computational cost. The neck is designed to aggregate low-level physical features with high-level semantic features, and then construct a pyramid at each level of the feature map. The head consists of multiple convolutional layers and predicts the final detection result based on the multilevel features in the neck section. From a structural point of view, it can be categorized into anchor-based and anchor-free, or more precisely, into a parameter-coupled head and parameter-decoupled head. In YOLOv6, two scaled parameterizable backbones and necks are proposed to accommodate models of different sizes, in addition to a decoupled head with a hybrid channel strategy.

#### 3.1.1. Backbone Network

RepVGG [21] proposes a structure reparameterization method. It uses a multi-branch structure for training and a single-branch structure for inference to obtain a better trade-off between speed and accuracy. Inspired by this, YOLOv6 designs an efficient backbone called EfficientRep [18]. Its specific design structure is shown in Figure 2. Its main component is RepBlock in the training phase, as shown in Figure 3. During the inference phase, each Repblock is converted into a stack of 3 × 3 convolutional layers, which has a ReLU activation function, denoted as RepConv, as shown in Figure 4. Thanks to the highly optimized 3 × 3 convolutional layers on mainstream CPUs and GPUs, the backbone network can fully utilize the computational power of the hardware, enhancing the representation capability while reducing the inference latency.

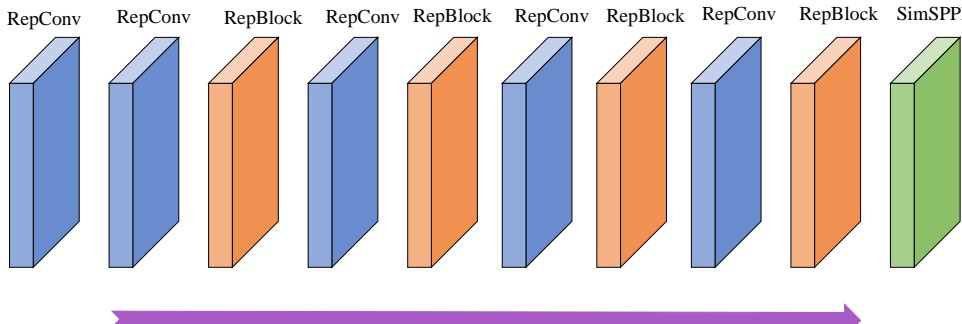

**Figure 2.** Framework for EfficientRep [21].

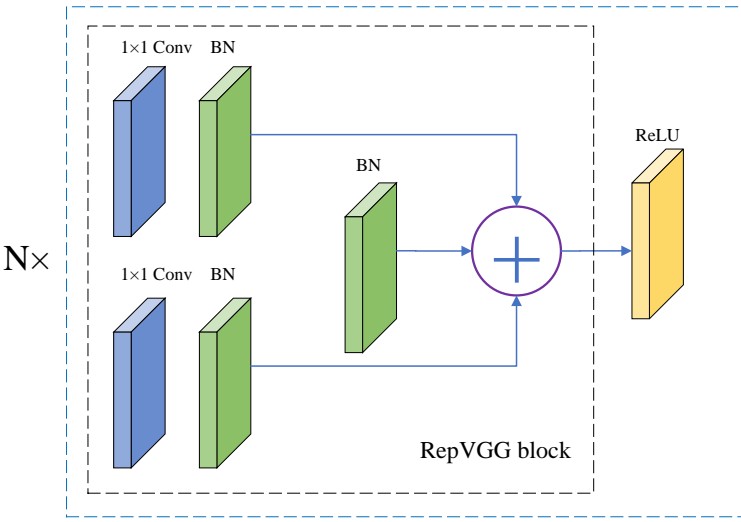

**Figure 3.** Framework for RepBlock [21].

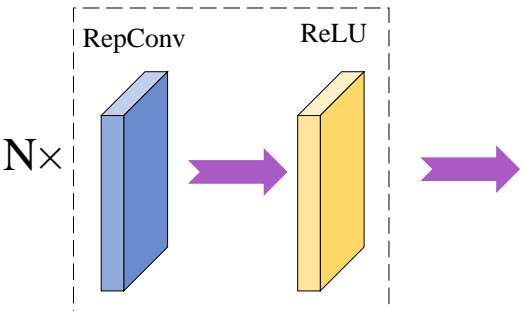

**Figure 4.** Framework for RepConv [21].

### 3.1.2. Neck Network

YOLOv6 uses the PAN structure of YOLOv5 as a basis for improvement. The CSPBlock in YOLOv5 is replaced with RepBlock (for small models) and CSPStackRep Block (for large models), and the depth and width are adjusted appropriately. The improved neck is named Rep-PAN [18], shown in Figure 5.

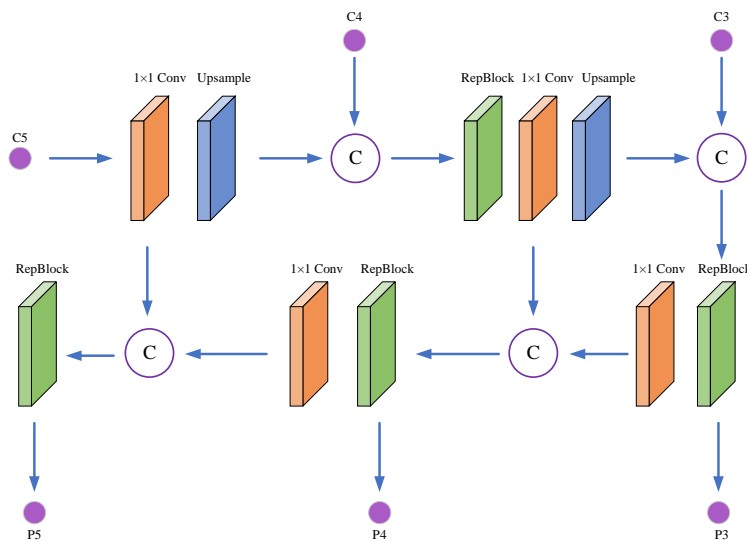

**Figure 5.** Framework for Rep-PAN [18]. C3, C4, and C5 are inputs, P3, P4, and P5 are outputs, and C stands for concat operation [21].

### 3.1.3. Head Network

　　YOLOv6 employs a hybrid channel strategy that retains only one $3 \times 3$ convolutional layer as a means of building an efficient decoupled head. The width of the head is then co-scaled by the multiplier of the widths of the backbone and neck. With these improvements, the inference delay is further reduced.

### 3.2. Effective Attention Mechanisms

　　The CBAM [22] attention mechanism is an attention module for feed-forward convolutional neural networks. CBAM contains two separate modules, the channel attention module (CAM) and spatial attention module (SAM). The framework diagram is shown in Figure 6. Given an intermediate feature map, these two modules sequentially infer the attention map and then multiply the attention map by the input feature map to perform adaptive feature modification.

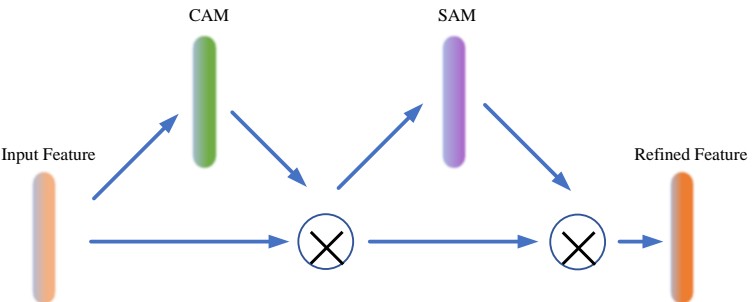

**Figure 6.** Framework for CBAM [21].

　　CAM first obtains two $1 \times 1 \times C$ feature maps by maximum pooling and average pooling. Then it feeds them into the *MLP*, and the features output from the *MLP* are summed at the pixel level before sigmoid activation to obtain the final channel attention features. Specifically, the input feature map $F(H \times W \times C)$ undergoes global maximum pooling and global average pooling to generate two $1 \times 1 \times C$ feature maps. The feature maps are then fed into a two-layer neural network (*MLP*). In the *MLP*, the number of neurons in the first layer is $C/r$ ($r$ is the reduction rate), and the number of neurons in the second layer is $C$. Then, the outputs of the feature from the *MLP* are subjected to an element-wise summation-based operation, which undergoes a sigmoid activation operation and generates the channel attention feature, i.e., $M_c$. Finally, the features $M_c$ and $F$ are subjected to an element-wise multiplication operation to generate the features required for SAM. The CAM framework diagram is shown in Figure 7. The equation is as follows:

$$M_c(F) = \sigma(MLP(A\gamma gPool(F)) + MLP(MaxPool(F))) = \sigma(W_c(W_0(F^c_{cog})) + W_c(W_0(F^c_{cog}))), \tag{1}$$

where $\sigma$ is the sigmoid function, $W_0 \in R^{C/r \times C}$, $W_1 \in R^{C \times C/r}$, and $W_0$ and $W_1$ are the two input weights of *MLP*.

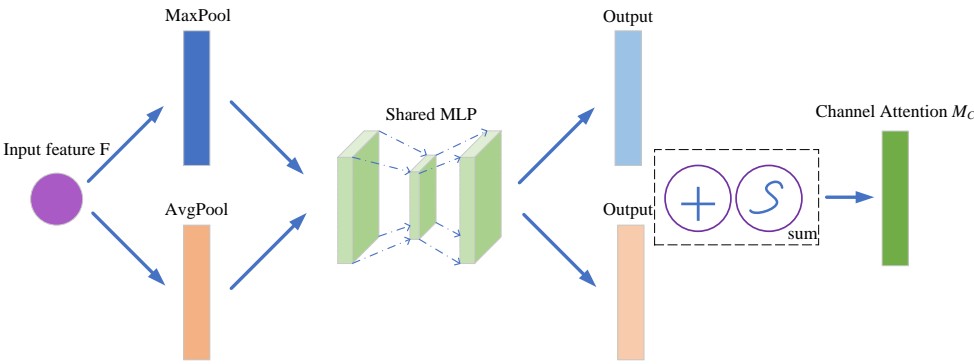

**Figure 7.** Framework for CAM [22].

SAM takes CAM as input and then performs maximum pooling and average pooling to finally obtain the $H \times W \times 1$ feature map. Subsequently, the feature map is channel-cascaded and downscaled to a single channel by a $7 \times 7$ convolutional layer. Finally, SAM is obtained after sigmoid activation. The framework of SAM is shown in Figure 8. The equation is as follows:

$$M_S(F) = \sigma(f^{7 \times 7}([AvgPool(F); MaxPool(F)])) = \sigma(f^{7 \times 7}([F_{avg}^S; F_{max}^S])), \tag{2}$$

where $\sigma$ is the sigmoid function and $f^{7 \times 7}$ represents the $7 \times 7$ convolutional layer operation.

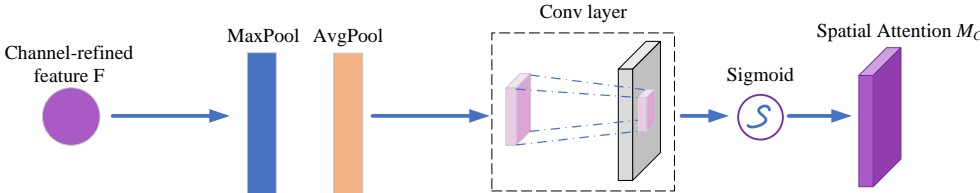

**Figure 8.** Framework for SAM [22].

*3.3. Suitable Loss Function*

In deep learning, training a model is the process of reducing the error by adjusting the model parameters. The loss function serves to measure the error between the model's predicted and actual results, and the parameters are adjusted by optimizing the loss function. The mean square deviation function is usually used in regression problems to calculate the square of the average difference between the predicted and actual values. The cross-entropy function is usually used in classification problems to calculate the difference between the predicted results and the actual labels. During training, models with low loss function values have more accurate predictions. GIoU used by YOLOv6 focuses not only on overlapping regions, but also on other non-overlapping regions. Compared to GIoU, DIoU [20] is more consistent with the mechanism of object frame regression. It takes the distance between the object and anchor, the overlap rate, and the scale into account, which makes the object frame regression more stable and does not have problems such as dispersion during the training process, like IoU and GIoU. Inspired by this, we changed the loss function to CIoU. CIoU is a measure of the accuracy of the object detection model, which mainly calculates the distance between the predicted value of the object box and the real marked box. Compared with the commonly used IoU metric, CIoU can more accurately measure the difference between the predicted box and the labeled box. CIoU considers the aspect ratio, which further improves the regression progress. The penalty term of CIoU is an influence factor added to the DIoU penalty that adapts the predicted box aspect ratio to the actual box aspect ratio. Its equation is as follows:

$$L_{CIoU} = 1 - IoU + \frac{\rho^2(b, b^{gt})}{C^2} + \partial v, \tag{3}$$

where $b$ and $b^{gt}$ represent the centers of the predicted and real frames, respectively. $\rho$ is the Euclidean distance between the two center points. $C$ represents the diagonal distance of the smallest closure region that can contain both the prediction frame and the real frame. $\partial$ is the weight function. $v$ is used to measure the similarity of the aspect ratios.

## 4. Experiment

In Section 4, we begin by describing the experimental configurations. We then compare the performance of the different YOLO series on the WFDS2000. Eventually, we test our improved model and show the effect graphs.

### 4.1. Experimental Setup

This program is improved in pytorch(Meta AI Research, America) and the MMYOLO framework. And it is trained and tested on two Nvidia 3090, which are with 24 G of video memory. The dataset was divided in a 7:3 ratio, and the learning rate was 0.001. We trained each network for 200 epochs. We use the average lookup rate (AP) to evaluate the detection accuracy.

### 4.2. Method Comparison and Visualization

We selected six models, YOLOv5, YOLOv6, YOLOv7, YOLOv8, YOLOX, and our proposed improved version of YOLOv6, and tested them on WFDS2000. We counted mAP 0.5, the amount of computation (FLOPs), the number of parameters (Params), and the computation speed (FPS) as evaluation metrics. The specific data are shown in Table 2.

**Table 2.** Performance of different models on the dataset.

| Method | FLOPs | Params | FPS | $AP_L$ | $AP_M$ | $AP_S$ | mAP |
|--------|-------|--------|-----|--------|--------|--------|-----|
| YOLOv5 | 53.975G | 46.144M | 30.64 | 0.329 | 0.223 | 0.091 | 0.548 |
| YOLOv6 | 21.882G | 17.188M | 33.9 | 0.396 | 0.232 | 0.105 | 0.592 |
| YOLOv7 | 51.749G | 36.508M | 10 | 0.308 | 0.211 | 0.082 | 0.547 |
| YOLOv8 | 82.557G | 43.631M | 46.1 | 0.409 | 0.245 | 0.111 | 0.598 |
| YOLOX | 77.659G | 54.149M | 42.3 | 0.368 | 0.211 | 0.068 | 0.586 |
| Ours | 21.883G | 17.23M | 32.5 | 0.421 | 0.241 | 0.106 | 0.619 |

Red indicators represent optimal data and blue indicators represent sub-optimal data.

For object detection, the average accuracy is the most important evaluation metric. Therefore, improving the average accuracy of the model is a top priority in model design. From Table 2, we can observe that before the improvement, the mAP of YOLOv8 is optimal and can reach 0.598. After our improved model is proposed, the mAP of ours is as high as 0.619, which is 0.021 higher than that of YOLOv8. For large objects, the improved AP, compared to the suboptimal YOLOv8, went up by 0.012. For small and medium-sized objects, the AP of the improved model is suboptimal. Overall, the AP of our model is optimal. The computational and parametric quantities of the model are important indicators of the model's complexity. In the current study, lightweight models are more favored. The smaller the computational and parametric quantities of the model, the more lightweight the model is. YOLOv6 has the smallest FLOPs and Params among a group of models, and it can be said that it is the most lightweight model. This is a theoretical basis for us to choose YOLOv6 for improvement. In our improved model, the parameters and computation of the model are slightly increased due to the addition of the attention mechanism. Compared with the original model, the FLOPs increased by 0.001G and the Params increased by 0.042 M. In addition, FPS is an important measure of model performance and response computational cost. The faster the computing speed, the smaller the cost of time spent. Among the models, YOLOv8 has the fastest computing speed, followed by YOLOX. YOLOv6 ranks third. Affected by the increase in FLOPs and Params of the model, the computational speed of the improved model decreases by 0.6. The performance of the computing speed is a pity for our model, whose time cost is higher.

However, our improved model trades a very small increase in FLOPs and Params and a decrease in FPS for a significant increase in mAP, which is a smaller cost in exchange for a larger boost. Accordingly, we have grounds to believe that our improvements to the model are effective.

To visualize the results of our experiments, we selected some of the effect diagrams, as shown in Figures 9–12.

In Figure 9, we show the effect of object detection in an outdoor environment. The large area of smoke did not affect our detection of the flames in it. As can be seen from the figure, for small smoke with distinctive features, we generate additional small detection frames. This can effectively minimize the cases of missing detection.

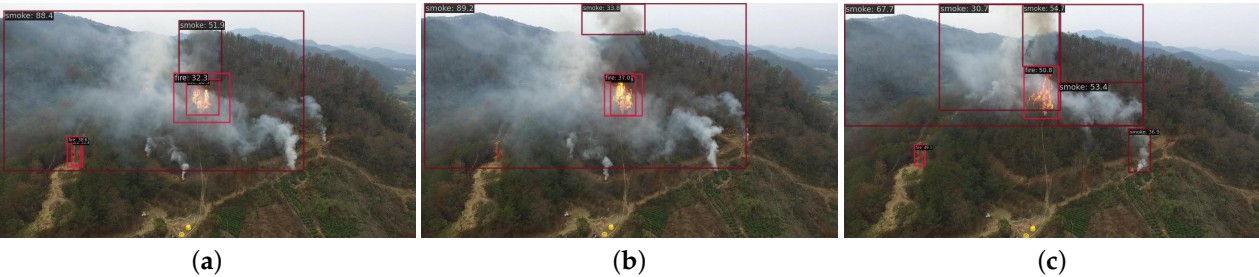

| (**a**) | (**b**) | (**c**) |

**Figure 9.** Visualization results of our method on WFDS2000 for outdoor environment. (**a**–**c**) are the detection results for different scenes.

Since there are many aerial images in the current object detection datasets, to better validate the model's detection effect on aerial images, we added some aerial images to our test. In Figure 10, the small area of fire in the aerial image can be recognized perfectly. This also further verifies the detection performance of our method on aerial images.

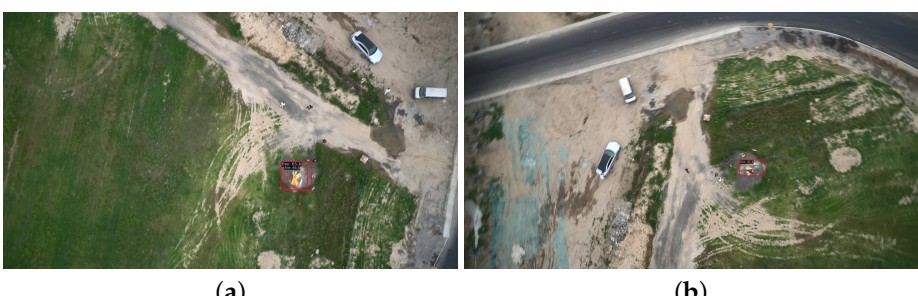

| (**a**) | (**b**) |

**Figure 10.** Visualization results of our method on WFDS2000 for aerial images. (**a**,**b**) are the detection results for different scenes.

The color of smoke varies from light to dark depending on the burning material and degree. To better verify the sensitivity of our model to smoke, we selected different colors of smoke. In Figure 11, there are four different degrees of smoke, varying from light to dark. From the figure, we can see that our model can accurately recognize these four degrees of smoke, and the darker the smoke color, the better the recognition performance.

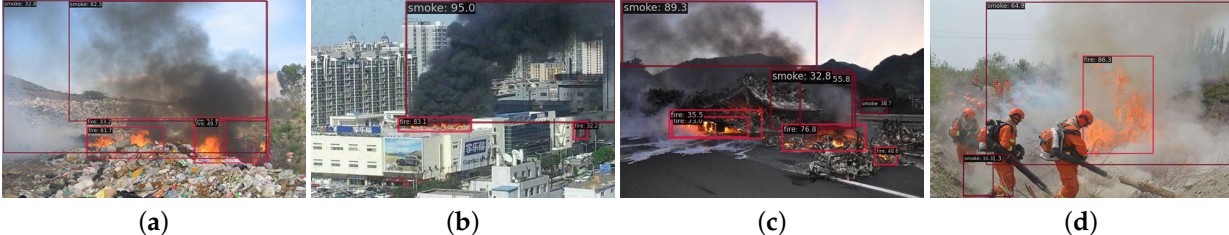

**Figure 11.** Visualization results of our method on WFDS2000 for different colors of smoke. (**a**–**d**) are the detection results for different scenes.

The weather factor is also an important factor affecting fire and smoke detection. We show how well the model detects smoke in a nighttime environment in Figure 12. As we can see from the figure, our model can clearly identify the fire. For smoke, we can recognize the smoke at the edge of the fire. For the smoke that lacks part of the light source, we cannot extract enough features due to the influence of the environment. Therefore, the recognition result of this part is not very satisfactory.

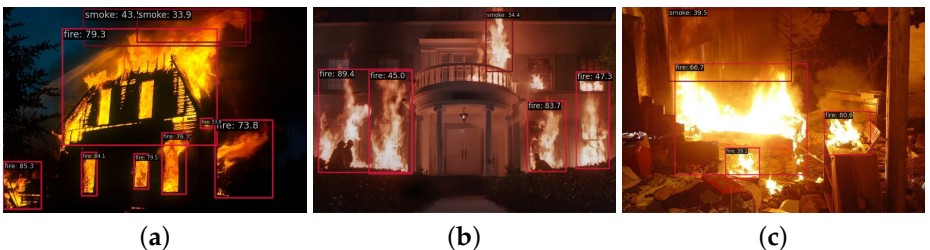

**Figure 12.** Visualization results of our method on WFDS2000 for night environment (**a**–**c**) are the detection results for different scenes.

## 5. Current Challenges and Future Directions

There are still many problems and challenges in the field of smoke and fire object detection. On the one hand, compared with natural images, fire and smoke images have problems, such as diverse application scenarios and directions, which are difficult to locate. On the other hand, outside circumstances, like weather and lighting, can also have an impact on the image. Therefore, we will pay more attention to feature extraction, lightweight network design, and sample scarcity of the dataset in the following.

### 5.1. Feature Extraction

An essential aspect is feature extraction. An efficient feature extraction method helps the model to obtain as many object features as possible, which will effectively enhance the detection of small and occluded objects. Recently, many excellent models have provided us with many creative feature extraction methods. Efficient RepGFPN [23], proposed by DAMO-YOLO [23], has excellent multi-scale feature fusion capability. The creative log2(n) layer-hopping connectivity in GFPN provides more efficient information transfer, which allows the feature information to be extended to a deeper network. In addition, it replaces the classical convolutional layer with rotational convolution, which helps to better extract object features with irregular orientations. SuperYOLO [24] better extracts features by processing images. It proposes the combination of RGB and infrared images, which can effectively utilize the advantages of each of the two images to minimize the missing object problem as much as possible. In addition, this method is very effective in tasks where small-scale objects and multi-scale objects coexist.

### 5.2. Lightweight Network Framework

The production of more productive and lighter-weight networks remains a hot research topic. It is necessary to develop more focused, lightweight, and productive object

detection network designs to improve detection performance. For example, Guan et al. [25] proposed a regionalized efficient network to suggestion generation and object classification, respectively. Its framework is designed to generate high-quality suggestions and then import the suggestions and input images into the network in order to learn convolutional features. Ye et al. [26] proposed an end-to-end multi-scale lightweight object detection network, the LSL-Net network, including LSM, EFM, and ADM. LSM extracts stable low-level information and reduces the overall computational effort. EFM extracts more efficient information to perform more accurate detection. ADM achieves higher accuracy for low-level objects at different scales, especially small objects.

*5.3. Datasets*

Currently, compared to other object detection tasks, there is a scarcity of datasets for the firework detection task, which can seriously hinder the development of firework detection methods. Therefore, emphasis should be placed on solving the problem of scarcity of dataset samples. In the production of the dataset, it should be combined with the existing image collection technologies, like LiDAR and GIS. Rich sample information not only advances pyrotechnic detection technology, but also provides credible support for actual applications.

*5.4. Future Directions*

Object detection methods should not only focus on their depth, but also on their breadth. Similarly, smoke and fire detection should not only be applied to forest fires, but it is also important for multi-scenario tasks, for example, fire monitoring of cities and fire monitoring of indoor environments. We sincerely hope that smoke and fire detection methods can be widely applied to multi-scenario tasks. Minimizing the impact of fire on people's lives and property is the point of our research.

## 6. Conclusions

This paper proposes an improved version of YOLOv6. We innovatively add the CBAM attention mechanism and CIoU loss function. CBAM can effectively improve the inference ability and multi-scale feature fusion ability. CIoU can better utilize the aspect ratio in the bbox regression triad. In terms of the dataset, we collected and labeled about 2000 multi-scene firework images. To validate the performance of our proposed model, we tested five methods of the YOLO series as well as the improved version of the method on this dataset and counted four evaluation metrics. The performance of our improved method was finally verified. However, from these evaluations, the object detection methods for smoke and fire detection still have much room for improvement. In our future work, we will enhance the feature extraction capability for smoke and fire. In addition, we will also focus on making the network more lightweight to reduce the number of parameters and computations of the model as much as possible. Due to the small dataset of smoke and fire detection, the sample-less model is also a future research focus of ours. In the future, we will also try to apply new models to smoke and fire detection and keep exploring this research area.

**Author Contributions:** Conceptualization, X.W. and Y.S.; software, A.W.; investigation, A.W.; experiment, A.W. and G.L.; formal analysis, X.W.; writing—original draft preparation, X.W. and A.W.; writing—review and editing, X.W. and Y.S.; supervision, Y.S. and X.W.; funding acquisition, Y.S. All authors have read and agreed to the published version of the manuscript.

**Funding:** This research was funded by the Natural Science Foundation of Shandong Province (ZR2022QF037).

**Institutional Review Board Statement:** Not applicable.

**Informed Consent Statement:** Not applicable.

**Data Availability Statement:** The data presented in this study are available on request from the corresponding author.

**Acknowledgments:** We would like to thank anonymous reviewers for their supportive comments that improved our manuscript.

**Conflicts of Interest:** The authors declare no conflict of interest.

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
