# Peer review of "Application of the YOLOv6 Combining CBAM and CIoU in Forest Fire and Smoke Detection"

_forests, doi:10.3390/f14112261_

Round 1
Reviewer 1 Report
Comments and Suggestions for Authors
This paper innovatively employs the CBAM attention module based on YOLOv6 to increase the extraction of smoke and fire features. The SIoU loss function is replaced with the CIoU loss function to ensure that the training time is reduced while extracting feature effects. Automatic mixed-accuracy training is used to train the model. The proposed model was tested on dataset containing multiple scenes. The experiments show that the model has a high response speed and accuracy in real-field smoke and fire detection. However, from these evaluations in the paper, the object detection methods for smoke and fire detection still have much room for improvement.
It is necessary to improve the design of the illustrative part of the work associated with drawings. For example, the photographs in Figure 1 are not marked alone and is there a need for so many photographs?
Author Response
Dear reviewer, we are very grateful for your support of our paper. Your recognition means a lot to us. In addition, your valuable comments are also very important to us. We hope that our next additions and revisions will give you a better impression of our thesis. Once again, we would like to express our sincere gratitude to you.
Response:
“It is necessary to improve the design of the illustrative part of the work associated with drawings. For example, the photographs in Figure 1 are not marked alone and is there a need for so many photographs?”
Answer: Our dataset contains multi-scene images of smoke and fire. In order to better present our dataset to the reader, so we have selected six images of WFDS2000 from different scenarios for presentation. Based on your comments, we have filtered the illustrations in Figure 1 and renamed it. Moreover, we have relabeled the images individually in Figure 1 as follows.
Please review the attached.

Reviewer 2 Report
Comments and Suggestions for Authors
This paper discusses the challenges of detecting forest fires and smoke in open environments and proposes an improved version of YOLOv6 for smoke and fire object detection. It introduces the CBAM attention module, CIoU loss function, and automatic mixed-accuracy training. The model is tested on a self-built dataset and is found to have high response speed and accuracy in real-field smoke and fire detection.
Strengths:
- The paper is well-written, structured, and easy to read.
- This paper innovatively uses YOLOv6 as the benchmark smoke and fire detection model, providing an alternative approach for researchers to evaluate performance.
- Incorporating the CBAM attention mechanism, CIoU loss function, and automatic mixed-accuracy training showcases a thoughtful effort to enhance the model's inference ability, accuracy, and hardware efficiency.
Weaknesses:
- While this paper introduces improvements to the model, it lacks a thorough comparison with existing state-of-the-art models or techniques, making it challenging to assess the significance of these enhancements.
- This paper mentions using a self-built dataset but provides limited details about its size, diversity, and quality. More information
- This paper mentions testing five methods of the YOLO series and the improved version of the dataset but does not provide specific results or metrics, making it difficult to gauge the model's performance objectively.
- The discussion of future work lacks specificity and actionable plans for enhancing feature extraction, lightweight the network, or addressing sample scarcity. Clear research directions would enhance the paper's value.
- While this paper highlights the importance of smoke and fire detection, it could benefit from discussing potential real-world applications and deployment scenarios beyond forest fires to showcase the broader significance of the research.
- Would you please add and discuss some of the mentioned references? What is the advantage/drawback of the presented approach?
- Fine-tuning the parameters are not discussed at all.
- Lack of computational cost analysis. According to the implementation details and the overall inference process (testing time), the authors are suggested to conduct some analysis on this issue.
Author Response
We are grateful to the noteworthy honorable reviewer for valuable recommendations and remarks. Your serious and rigorous judging attitude makes us unforgettable. We welcome the time and efforts the reviewers spent on our submitted manuscript. Underneath, we accomplish the suggestions.
Response:
- “While this paper introduces improvements to the model, it lacks a thorough comparison with existing state-of-the-art models or techniques, making it challenging to assess the significance of these enhancements.”
Answer: In this paper, we select five comparison methods, among which, YOLOv5 is one of the most widely used and portable object detection methods.YOLOv6, YOLOv7 and YOLOv8 are released in 2022 and 2023, which can represent the latest cutting-edge algorithms in the current YOLO series. Therefore, we select these five methods for comparison. In addition, in order to make our comparison more comprehensive, we did not compare mAP alone. In contrast, we comprehensively compared four evaluation metrics, namely, Params, FPS, FLOPs, mAP, APL,APs, and APM. Finally, we synthesize the advantages of the methods in this paper from these seven aspects.
- ”This paper mentions using a self-built dataset but provides limited details about its size, diversity, and quality. More information ”
Answer: Inspired by your comments, we have made relevant additions. We give additional description of the dataset in lines 118 through 119 and show the specific data in Table 1. The specific descriptions are as follows:
In Table 1, we present the details of the dataset. It contains the number of images and the specific number of pyrotechnic objects in the training set, validation set and test set.
- “This paper mentions testing five methods of the YOLO series and the improved version of the dataset but does not provide specific results or metrics, making it difficult to gauge the model's performance objectively.”
Answer: Dear reviewers, we provide detailed data for 7 metrics in Table 2. We calculated seven evaluation metrics including Params, FLOPs, FPS, mAP, APL, APs and APM to evaluate and illustrate the performance of different models. In addition, we fully discuss and analyze the experimental metrics in rows 228 to 252, and the experimental results prove that the performance of our improved model is better.
- “The discussion of future work lacks specificity and actionable plans for enhancing feature extraction, lightweight the network, or addressing sample scarcity. Clear research directions would enhance the paper's value.”
Answer: Based on your suggestion, we have described it in detail in Section 5. Related issues are discussed in subsections 5.1, 5.2 and 5.3.
- ” While this paper highlights the importance of smoke and fire detection, it could benefit from discussing potential real-world applications and deployment scenarios beyond forest fires to showcase the broader significance of the research.”
Answer: Based on your suggestions, we further discussed practical application scenarios for this study. We revisit the significance of this research by prospecting future application areas and research value in Subsection 5.4. The specific descriptions are as follows:
Object detection methods should not only focus on their depth, but also on their breadth. Similarly, smoke and fire detection should not only be applied to forest fires, but it is also important for multi-scenario tasks. For example, fire monitoring of cities and fire monitoring of indoor environments. We sincerely hope that smoke and fire detection methods can be widely applied to multi-scenario tasks. Minimizing the impact of fire on people's lives and property is the point of our research.
- “Would you please add and discuss some of the mentioned references? What is the advantage/drawback of the presented approach?”
Answer: We discuss some of the latest excellent methods in Section 5 and add some references. In addition, we discuss the advantages and disadvantages of the methods in this paper in lines 249 to 252. The specific descriptions are as follows:
Advantages:
However, our improved model trades a very small increase in FLOPs and Params and a decrease in FPS for a significant increase in mAP. A smaller cost in exchange for getting a larger boost. Accordingly, we have grounds to believe that our improvements to the model are effective.
Disadvantages:
The performance of the computing speed is a pity for our model, whose time cost is higher.
- ” Fine-tuning the parameters are not discussed at all.”
Answer: For the fine-tuning of the experimental parameters, we have a specific description in lines 219 through 222. Regarding the selection of experimental training parameters, we use the model weights corresponding to the lowest loss function from 200 epochs of server training. No additional parameter fine-tuning is required, and only the best parameters obtained from deep learning are used.
- ” Lack of computational cost analysis. According to the implementation details and the overall inference process (testing time), the authors are suggested to conduct some analysis on this issue.”
Answer:For the analysis of computed costs, we have added lines 243 through 246. The specific descriptions are as follows:
In addition, FPS is an important measure of model performance and response computational cost. The faster the computing speed, the smaller the cost of time spent. Among the models, YOLOv8 has the fastest computing speed, followed by YOLOX. YOLOv6 ranks 3rd. Affected by the increase in FLOPs and Params of the model, the computational speed of the improved model decreases by 0.6.
Please review the attached.
